



# Implementing a finite-volume coupled physical-biogeochemical model to the coastal East China Sea

Jingui Liu[1, 2*], Shanglu Li[3], Xuanliang Ji[1, 2], Guimei Liu[1, 2], Qingqing Pan[1], Yun Li[1, 2]

[1] National Marine Environmental Forecasting Center (NMEFC), Minister of National Resources (MNR), Beijing, 100081, China

[2] Key Laboratory of Marine Hazards Forecasting, NMEFC, MNR, Beijing, 100081, China

[3] Marine Monitoring and Forecasting Center of Zhejiang, Hangzhou, 310007, China

*Correspondence to:* Jingui Liu (lehel1104@yahoo.com)

**Abstract:** Several models for estuarine physical processes and biogeochemistry have been developed over last decades. One of the most comprehensive coupled model systems, Finite Volume Community Coastal Model (FVCOM) coupled with European Regional Seas Ecosystem Model (ERSEM) through the Framework for Aquatic Biogeochemical Models (FABM) has been implemented to a high resolution coastal East China Sea (ECS), which encompassed complex coastal zone and part of continental shelf. Physical model was assessed by traditional univariate comparisons, while a rigorous model skill assessment was conducted for coupled biological model. The model system's ability to reproduce major characteristics both in physical and biological environments was evaluated. The roles of physical, chemical and environmental parameters on the biogeochemistry of the ECS were extensively studied. This work could form a significant basis for future work, e.g. the response of biogeochemical flux to physical mechanism.

**Keywords:** Biogeochemistry; coastal East China Sea; FVCOM; ERSEM; FABM



## 1 Introduction

Extensive ocean models have been developed over last several decades to serve as tools for research and maritime projects. A demand for explicit modeling of combined physical, chemical and biological systems begins on a growing realization that biogeochemical state cannot be inferred from their physical properties alone (Blackford et al., 2004; Tomasz et al., 2014). Traditional numerical models have been constructed based on simplified assumptions on the functionality of complex marine ecosystem. Most of them failed to simulate important biogeochemical processes, because the models did not consider essential features, such as explicit carbon cycling, microbial food dynamics, the role of key functional groups and multiple nutrient limitation to primary production (Mateus et al., 2012). In recent years, marine ecosystem models have been explored to understand, quantify and estimate biogeochemical processes in seas and oceans. These models vary in complexity from simple four-compartment Nitrate, Phytoplankton, Zooplankton, Detritus (NPZD) pelagic models (Oschlies et al., 2000; Dabrowski et al., 2013) to more complex multi-functional group models describing ocean biogeochemistry and lower trophic food web (Mateus, 2012; Flynn, 2010; Follows et al., 2007; Wild-Allen et al., 2010). European Regional Seas Ecosystem Model (ERSEM) is one of the most established and complex ecosystem models for lower trophic levels of marine food web in use, which assesses over 40 state variables with benthic-pelagic process (Baretta et al., 1995). The model has been applied in a wide number of contexts that included short-term forecasting (Edwards et al., 2012), ocean acidification (Blackford and Gilbert, 2007), coupled climate-acidification projections (Polimene et al., 2014), and biogeochemical cycling (Wakelin et al., 2012).

The East China Sea (ECS), a marginal sea of western North Pacific Ocean, is characterized by wide shelf, complex circulations, and fresh water inputs from the Yangtze River (YR), the Qiangtangjiang River (QR), the Oujiang River (OR), and the Minjiang River (MR). Land-source nutrient flux interacts with ocean current, making the cycle of nutrients richer and more complicated. In recently years, with rapid development of coastal ocean economy, tremendous amounts of pollutants, such as nitrate and phosphorus, have been exported into coastal oceans due to an increase in anthropogenic activity. Consequently, the structure and function of ecosystem may be affected (Wang et al., 2004). Eutrophication has been considered as the most serious environmental problem of coastal ECS, e.g. harmful algae blooms (mainly red tide), which greatly impacts on human health, aquatic ecosystem, and



70 the economy. Previous research have focused on this ecosystem hazards by illustrating lower nutrient

food web along coastal region of ECS (Wang et al., 2006; Guo et al., 2014; Li et al., 2008; Ye et al.,

2015; Zhang et al., 2004). However, most of these studies were based on either limited field surveys or

simplified laboratory experiments. There were also defects in providing environmental drivers offline,

the simple zero-dimensional box biological model, and limited spatial and temporal coverage (Wang et

75 al., 2013; Zhu et al., 2005). Online three-dimensional physical and biogeochemical fluxes need to be

considered for more realistic representations.

In this study, we implemented the high-resolution FVCOM coupled to the ERSEM through the FABM

framework. The paper was organized as follows. Section 2 briefly described the study area and

observations used. Section 3 focused on the FVCOM-ERSEM model, and specific setup. Model skill

80 assessment was carried out in Section 4. A discussion was further explored in Section 5. Conclusions

were drawn and future work was discussed in Section 6.

**2 Study area and observations**

Study area covers the coast of ECS extending from southern Taiwan Strait to northern Yangtze River

(YR) water system. Model domain is 117 -124.5°E and 22-33°N within a 174 m-isobath (Fig.1).

85 Physical environment of ECS has a distinct seasonality feature at mid-latitude and influenced by

anthropogenic stresses from adjacent landmass, as well as mixing from several principal water types.

From the north to the south, the domain comprises several important estuaries, including: YR, QR river,

OR, Aojiang River (AR), Feiyunjiang River (FR), MR and Jiulongjiang River (JR) and empties into

ECS. The topography of model domain is derived from ETOPO1 (1-minute gridded data) for the open

90 ocean region (Amante and Eakins, 2009) and nautical chart for estuarine areas.

Monitored data were collected from 5 ecological buoys monitoring at the Zhejiang coast in May 2019,

including temperature, salinity, conductivity, pH, dissolved oxygen (DO), dissolved oxygen saturation,

turbidity, chlorophyll-a (Chla), phosphate, nitrate and ammonia. Meanwhile, tidal level observations

were also collected at several tidal stations to evaluate model performance.

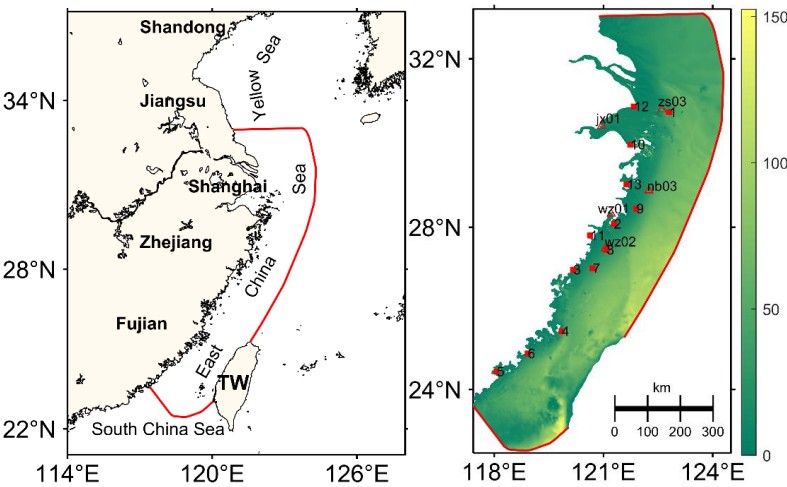

**Fig. 1.** Study area (left panel) and bathymetry map with locations of tidal stations and ecological buoys (right panel). Tidal stations are marked by square and ecological buoys by triangle. Tidal stations: Shengshan (SS) "1", Kanmen (KM) "2", Sansha (SS2) "3", Pingtan (PT) "4", Xiamen (XM) "5", Chongwu (CW) "6", Taishan (TS) "7", Nanji (NJ) "8", Dachendao (DI) "9", Zhenhai (ZH) "10", Ruian (RA) "11", Luchaogang (LP) "12", Sanmenjiantiao (S3) "13". Ecological buoys: zs03, jx01, nb03, wz01 and wz02.

## 3 Model description and setup

### 3.1 Hydrodynamic model

The numerical model used in this study is unstructured grid based, free-surface, 3-D primitive equations Finite-Volume Community Ocean Model (FVCOM) ocean model described in detail by Chen et al. (2003a). To date, current version is fully coupled ice-ocean-wave-sediment-ecosystem model system with options of various turbulent mixing schemes, generalized terrain-following coordinates and wet/dry treatments. Finite-volume approach combines finite-element method for geometric flexibility and finite-difference method for simple discrete structures, in order to enhance the computational efficiency. Multiple dynamical forces, including river runoff, astronomical tide, waves, mean flow, wind, etc. and seasonal temperature, salinity and density, coexist and interact in study area. Therefore, unstructured, finite-volume ocean model can fit to ECS situation sensationally.

### 3.2 Biogeochemical model

ERSEM was developed in the 1990's, which is one of the most established ecosystem models for the lower trophic levels of the marine food web (Butenschön et al., 2016). The current model release



contains the essential elements for the pelagic and benthic parts of the marine ecosystem, including the microbial food web, the carbonate system, and calcification.

Trophic structure is defined on the basis of a predatory action of consumers on producers, bacteria, and themselves. A benthic module is implemented to estimate the mineralization of sinking organic matter, nutrients and oxygen flux at bed-water interface. A pelagic model comprises more than twenty-two

major state variables: light, producers, consumers, decomposers, pelagic organic matter (dissolved labile, semi-labile and semi-refractory DOM, small-size, medium-size and large-size POM), benthic organic matter (dissolved, particulate and refractory), nutrients (nitrate, ammonium, phosphate and silicate), and oxygen. For this simulation, we have considered four types of primary producers: diatoms, nanophytoplankton, picophytoplankton and microphytoplankton, and three group of consumers:

mesozooplankton, microzooplankton, and Heterotrophic flagellates.

The FABM enables complex biogeochemical models for marine and freshwater systems to be developed as sets of stand-alone or process specific modules (Bruggeman & Bolding, 2014). It has been coupled to many hydrodynamic models including GOTM (General Ocean Turbulence Model), ROMS (Regional Ocean Model System), NEMO (Nucleus for European Modelling of the Ocean), MOM (Modular Ocean

Model), HYCOM (HYbrid Coordinate Ocean Model), FVCOM and SCHISM (https://sourceforge.net/projects/fabm/).

Parameter values for each generic type model were listed in Tables 2 to 4. Whenever possible they have been adopted from original study performed in similar complexity (Butenschön et al., 2016). Some parameters like maximum specific productivity, mimimal specific lysis rate, assimilation efficiency of

mesozooplankton were estimated from references of ECS (Guo et al., 2014; Li et al., 2008; Gin et al., 1998; Wang et al., 2006).

### 3.3 Model configuration

The computational domain was divided into 102, 688 non-topped triangular cells with 53, 512 grid nodes. The resolution at open boundary was set up to 15 km, and refined to ~200 m around riverine

channel.

The model was forced by realistic tide, river discharge and atmospheric conditions. The tidal forcing was imposed using the TPXO7.2 data (Egbert and Erofeeva, 2002), which provides 8 primary harmonic constitute to predict ocean tide. Inputs for fresh water were prescribed using climatological monthly





values for respective main channels (Fig. 2), derived from China Water & Power Press

(http://www.waterpub.com.cn/). Riverine inputs of salinity were set to 0 psu. The temperature and

suspended sediment concentration were collected according to multi-year averaged monthly data sets

(Editorial Board of China Bay Survey). The surface wind forcing, heat flux and

precipitation/evaporation were acquired from 6-hour Reanalysis data of NOAA/s National Centers for

Environmental Prediction (NCEP) (ftp://ftp.cdc.noaa.gov/Datasets/ncep.reanalysis2/gaussian_grid).

Initial conditions for the temperature and salinity were derived from GDEM (Generalized Digital

Environmental Model). Open boundary conditions for temperature and salinity were extracted from an

ocean reanalysis data set of SODA (Simple Ocean Data Assimilation).

Surface boundary conditions were prescribed as no-flux for all biogeochemical state variables. Monthly-

averaged nutrients were imposed at riverine boundary of YR in Table 1, including nitrate nitrogen,

ammonia nitrogen, phosphorus, silicate (Wang, et al., 2013; Liu, 2002; Xu, 2019). Yearly- averaged

nutrients were specified at other riverine boundaries. The Nitrate nitrogen, ammonia nitrogen,

phosphorus, silicate concentrations were set as 80.4, 2.26, 1.53 and 120 μmol l-1 respectively at QR;

the corresponding concentrations were set as 20, 5.5, 1 and 150 μmol l$^{-1}$ at OR, AR and FY; and 53.3,

13, 0.5 and 221 μmol l$^{-1}$ at JR. Initial and open boundary conditions for the ecological model properties

(phosphate, nitrate, oxygen and silicate) were derived from WOA09 (World Ocean Atlas 2009), and the

Chla was from OC_CCI (Ocean Colour Climate Change Initiative). The initial values of ammonium

were given as a homogeneous constant value, 1.0 mmol N m$^{-3}$.

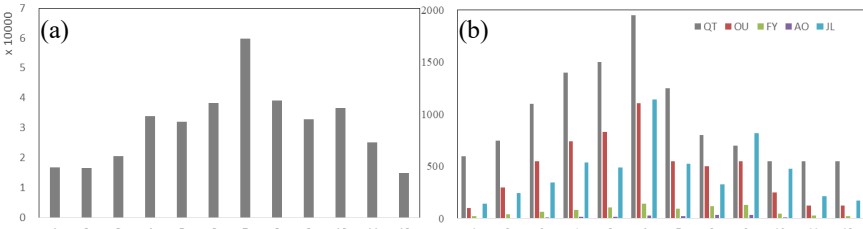

**Fig. 2. Multi-year monthly averaged river discharge: Datong station of Yangtze River (YR) (a); Qiangtangjiang River**
**(QR), Oujiang River (OR), Feiyunjiang River (FR), Aojiang river (AR), and Jiulongjiang river (JR) (b) (m3 s-1).**

An initial period of 1 month, January 2018, was used as a spin-off period for tide in barotropic mode,

and the model was run for 11 months until December in baroclinic mode with an initial tide fields. The

biogeochemical coupled model was run for a period of eight months, starting in December 2018 and



finishing in July 2019 which covers the period of harmful algal blooms, with an external time step of

1.5 s for the numerical simulations.

**Table 1. Monthly nutrients input of Yangtze River (µmol l-1).**

|        | Jan. | Feb. | Mar. | Apr. | May  | June | July | Aug. | Sep. | Oct. | Nov. | Dec. |
|--------|------|------|------|------|------|------|------|------|------|------|------|------|
| NH4    | 6.1  | 7.2  | 7.7  | 7.7  | 7.1  | 7.1  | 7.7  | 7.1  | 6.6  | 6.6  | 6.6  | 8.4  |
| NO3    | 103  | 127  | 127  | 127  | 100  | 100  | 100  | 86   | 86   | 86   | 103  | 103  |
| PO4    | 1.68 | 1.33 | 1.33 | 1.33 | 1.59 | 1.59 | 1.59 | 1.38 | 1.38 | 1.38 | 1.68 | 1.68 |
| SI     | 120  | 96   | 96   | 96   | 114  | 114  | 114  | 129  | 129  | 129  | 120  | 120  |

**Table 2. List of parameter values used as reference for primary producers: diatoms (P1), nanophytoplankton (P2), picophytoplankton (P3), and microphytoplankton (P4). All values are from Butenschön et al. (2016), except a from Guo et al. (2014), b from Li et al. (2008), c from Gin et al. (1998), and d Wang et al. (2006).**

| Parameter | Unit | P1 | P2 | P3 | P4 |
|-----------|------|----|----|----|----|
| Max. specific productivity at reference temperature | d$^{-1}$ | 3.7[b] | 1.625 | 0.9[a] | 2.0[b] |
| Q10 temperature coefficient | - | 2.0 | 2.0 | 2.0 | 2.0 |
| Specific rest respiration at reference temperature | d$^{-1}$ | 0.04 | 0.04 | 0.045 | 0.035 |
| Excreted fraction of primary production | - | 0.2 | 0.2 | 0.2 | 0.2 |
| Respired fraction of primary production | - | 0.2 | 0.2 | 0.2 | 0.2 |
| Minimum nitrogen to carbon ratio | mmol P (mg C)$^{-1}$ | 0.0042 | 0.005 | 0.006 | 0.0042 |
| Minimum P:C ratio | mmol N (mg C)$^{-1}$ | 0.0001 | 0.00023 | 0.00035 | 0.0001 |
| Maximum P:C (relative to redfield ratio) | - | 2.0 | 2.0 | 1.5 | 2.7 |
| Maximum nitrogen :C (relative to redfield ratio) | - | 1.075 | 1.075 | 1.05 | 1.1 |
| nitrate affinity | m$^3$ mg$^{-1}$ C d$^{-1}$ | 0.0004[d] | 0.004 | 0.006 | 0.0004[d] |
| ammonia affinity | m$^3$ mg$^{-1}$ C d$^{-1}$ | 0.0025 | 0.004 | 0.007 | 0.002 |
| phosphate affinity | m$^3$ mg$^{-1}$ C d$^{-1}$ | 0.019[d] | 0.004 | 0.006 | 0.020[d] |
| Maximum silicate to carbon ratio | mmol Si (mg C)$^{-1}$ | 0.0118 | - | - | - |
| Michaelis-Menten constant for silicate limitation | mmol m$^{-3}$ | 0.02 | - | - | - |
| 1.1 of minimal specific lysis rate | d$^{-1}$ | 0.03[c] | 0.05 | 0.22[a] | 0.03[c] |
| Initial slope of PI-curve | mg C m$^2$ mg$^{-1}$ | 4.0 | 5.0 | 6.0 | 3.0 |
| Photoinhibition parameter | mg C m$^2$ mg$^{-1}$ | 0.07 | 0.1 | 0.12 | 0.06 |
| Max. effective chlorophyll to carbon photosynthesis ratio | mg Chl (g C)$^{-1}$ | 0.06 | 0.025 | 0.015 | 0.045 |

**Table 3. List of parameter values used as reference for bacteria. Values from Butenschön et al. (2016).**

| Parameter | Unit | Value |
|-----------|------|-------|
| M-M constant for oxygen limitation relative to saturation state | - | 0.31 |
| M-M constant for nitrogen limitation | mmol N m-3 | 0.5 |
| M-M constant for phosphorus limitation | mmol P m-3 | 0.1 |
| Q10 value | - | 2.0 |
| Efficient at high oxygen levels | - | 0.6 |
| Efficient at low oxygen levels | - | 0.2 |
| Specific rest respiration at reference temperature | d$^{-1}$ | 0.1 |
| Maximum phosphorus to carbon ratio | mmol P (mg C)$^{-1}$ | 0.0019 |
| Maximum nitrogen to carbon ratio | mmol N (mg C)$^{-1}$ | 0.0167 |





| Specific mortality at reference temperature | $d^{-1}$ | 0.05 |
| Max. specific uptake at reference temperature | $d^{-1}$ | 2.2 |
| Semi-refractory DOC in proportion to activity respiration | 1 | 0.3 |
| Turn-over of POM relative to DOM | $d^{-1}$ | 0.01 |

**Table 4. List of parameter values used as reference for zooplankton. Z4, Z5 and Z6 indicate mesozooplankton, microzooplankton and nanoflagellates, respectively. Values from Butenschön et al. (2016).**

| Parameter | Unit | Z4 | Z5 | Z6 |
|---|---|---|---|---|
| Q10 temperature coefficient | - | 2.0 | 2.0 | 2.0 |
| Assimilation efficiency | - | 0.75 | 0.5 | 0.4 |
| Fraction of unassimilated detritus not respired | | 0.9 | - | - |
| Specific basal mortality | $d^{-1}$ | 0.05 | 0.05 | 0.05 |
| Max. mortality due to oxygen limitation | $d^{-1}$ | 0.2 | 0.25 | 0.3 |
| Fraction of unassimilated prey that is excreted | - | 0.5 | 0.5 | 0.5 |
| Max. phosphorus to carbon ratio | mmol P $(mg\ C)^{-1}$ | 0.00079 | 0.001 | 0.001 |
| Max. nitrogen to carbon ratio | mmol N $(mg\ C)^{-1}$ | 0.0126 | 0.0167 | 0.0167 |
| Specific rest respiration at reference temperature | $d^{-1}$ | 0.015 | 0.02 | 0.025 |
| Max. specific uptake at reference temperature | $d^{-1}$ | 1.0 | 1.25 | 1.5 |
| Food preference for P3 | 1 | 0.0 | 0.15 | 0.25 |
| Food preference for P4 | 1 | 0.15 | 0.1 | - |
| Food preference for P2 | 1 | 0.05 | 0.15 | 0.15 |
| Food preference for P1 | 1 | 0.15 | 0.15 | - |
| Food preference for B1 | 1 | 0.0 | 0.1 | 0.45 |

**4 Model skill assessment**

This section concerned physical and biological response of the model during numerical period. Traditional univariate comparisons were used to assess physical model skill. Time-series comparisons of water level, sea surface temperature, and sea surface salinity were presented in Figs. 3-5, whereas Table 5 showed the comparisons of harmonic constants for 8 main astronomical constituents.

A rigorous model skill assessment was conducted for coupled biological model, thus the model's capabilities were tested and understood. Herein, additional approaches were explored to validate the complex biological model performance, e.g. the Percentage of Bias (PB) and the Adjusted Relative Mean Absolute Error (ARMAE). We also attempted multivariate comparison of the modeled and the observed using the Cost Function (CF) to minimize model-data misfit.



### 4.1 Physical fields

Graphically comparing the modeled with the observed could be a useful way to assess model performance. We plotted time-series water level, SST and surface salinity. To quantify the differences between data and modeled results, the root-mean-square (RMS) error ($e_{rms}$) and the correlation coefficient ($r^2$) were employed as major skill assessment index of physical fields to indicate average discrepancy (Taylor, 2000).

$$e_{rms} = \left[\frac{1}{N}\sum_{n=1}^{N}(M_n - O_n)^2\right]^{1/2} \tag{1}$$

$$r^2 = \frac{\frac{1}{N}\sum_{n-1}^{N}(M_n - \bar{M})(O_n - \bar{O})}{\sigma_M \sigma_O} \tag{2}$$

where $M$ and $O$ were for the modeled and the observed, respectively. $\bar{M}$ and $\bar{O}$ were mean values, and $\sigma_M$ and $\sigma_O$ were standard deviations of $M$ and $O$, respectively.

### 4.1.1 Tidal analysis

In coastal and estuarine environment, tidal current represented one of main forcing on biogeochemical dynamics. It was crucial to correctly simulate tide propagation along the coast that could be characterized by water surface elevation. We selected six tidal stations to validate temporal water level of May 2019 (Fig. 3). As shown in Fig.3, simulated water levels were in good agreement with observations over entire measurement period except for the RA and the PT. The correlation coefficients $r^2$ ranged from 0.94 to 0.99, and the root-mean-square (RMS) error $e_{rms}$ from 0.155 to 0.59 m. The R station lay near AR estuary, where the coastline and terrain have been modified due to human reclamation in past several years, thus it appeared that flood tide could not pump high enough. The PT located near eastern coast of the PT Island, where the resolution of the terrain was quite coarse. Harmonic analysis were also carried out using the T_Tide Toolkit with observed tidal level and computed water level. Harmonic constants of eight main tidal constituents were listed in Table 5. Generally, maximum deviation of the amplitude was less than ~10 cm, and phase difference was less than ~20 degree. Compared to tidal ranges of coastal areas, model and observational data are in good agreement with each other.

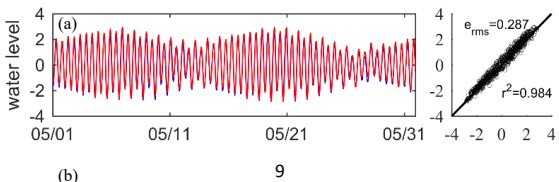

(b)





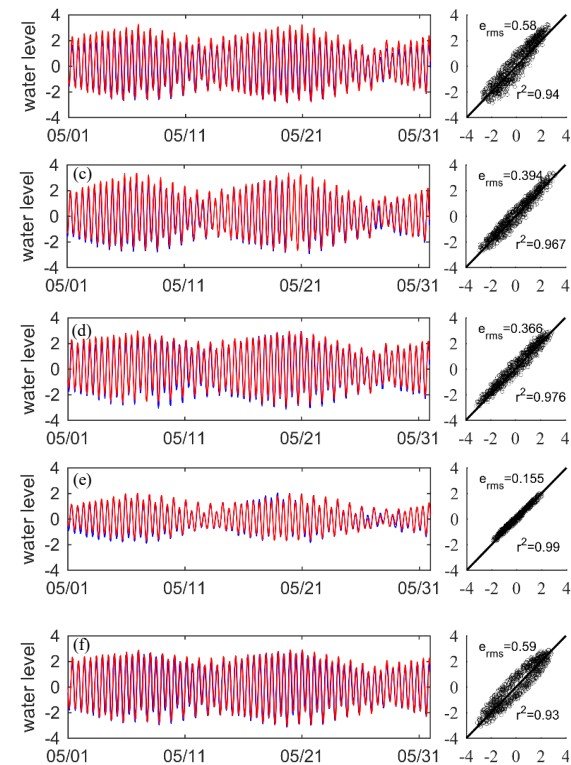

**Fig. 3. Validations of time series of water level during May 2019 at six tidal stations, including KM (a), RA (b), S3 (c),**

**SS2 (d), SS (e), and PT (f) (blue line for the observed and red line for the modeled).**

**Table 5. Harmonic constants of 8 main constituents**

| Cons. Stations | | M2 | N2 | S2 | K2 | Q1 | O1 | P1 | K1 |
|---|---|---|---|---|---|---|---|---|---|
| KM | Amplitude | 181.3 | 33.0 | 66.1 | 18.0 | 3.3 | 21.1 | 8.3 | 28.6 |
| | | 187.9 | 29.6 | 62.5 | 18.2 | 4.2 | 21.7 | 9.3 | 29.6 |
| | Phase | 251.2 | 228.6 | 292.0 | 291.5 | 164.1 | 180.2 | 215.5 | 216.9 |
| | | 260.6 | 238.9 | 303.5 | 299.6 | 162.9 | 186.0 | 219.8 | 219.7 |
| SS | Amplitude | 113.1 | 20.8 | 52.1 | 14.1 | 2.5 | 16.1 | 8.1 | 26.7 |
| | | 106.2 | 19.3 | 48.0 | 15.0 | 2.8 | 15.7 | 7.4 | 25.2 |
| | Phase | 278.4 | 262.7 | 320.9 | 319.8 | 140.2 | 152.6 | 191.5 | 191.4 |
| | | 273.6 | 257.9 | 323.5 | 320.5 | 132.3 | 153.5 | 194.5 | 193.1 |
| S3 | Amplitude | 179.5 | 32.3 | 74.6 | 19.3 | 4.1 | 22.2 | 9.6 | 30.6 |
| | | 185.7 | 29.1 | 69.2 | 20.6 | 3.5 | 21.3 | 8.7 | 29.2 |
| | Phase | 251.0 | 232.2 | 297.8 | 297.8 | 159.3 | 175.5 | 214.8 | 214.2 |
| | | 263.5 | 249.7 | 316.9 | 314.0 | 164.1 | 186.2 | 221.8 | 220.0 |
| SS2 | Amplitude | 202.7 | 38.0 | 67.2 | 18.6 | 4.9 | 24.3 | 10.0 | 30.9 |
| | | 202.2 | 31.3 | 56.8 | 16.2 | 4.6 | 22.9 | 9.6 | 230.1 |
| | Phase | 281.5 | 256.0 | 322.0 | 322.5 | 184.5 | 196.5 | 228.2 | 232.2 |



| | | | | | | | | | |
|---|---|---|---|---|---|---|---|---|---|
| DI | Amplitude | 290.3 | 263.9 | 330.6 | 325.9 | 174.7 | 193.5 | 30.8 | 231.0 |
| | | 139.8 | 28.0 | 60.7 | 16.0 | 6.7 | 19.0 | 11.3 | 32.4 |
| | | 145.1 | 25.1 | 55.8 | 16.5 | 3.8 | 21.3 | 8.9 | 28.4 |
| | Phase | 250.6 | 227.0 | 286.6 | 289.7 | 181.1 | 177.8 | 220.6 | 212.8 |
| | | 247.8 | 227.9 | 291.5 | 288.5 | 155.2 | 179.3 | 211.2 | 212.6 |
| PT | Amplitude | 206.3 | 37.8 | 62.8 | 17.4 | 4.9 | 25.2 | 9.9 | 31.0 |
| | | 214.7 | 32.3 | 52.1 | 15.0 | 4.8 | 24.4 | 9.7 | 31.2 |
| | Phase | 307.6 | 280.6 | 350.7 | 348.5 | 195.9 | 212.3 | 247.0 | 250.3 |
| | | 317.1 | 299.9 | 5.1 | 1.0 | 193.9 | 211.0 | 249.7 | 251.2 |

### 4.1.2 Surface sea salinity and temperature

Temperature structure was one of major limiting factor on primary productivity in coastal ECS. The temperature for prorocentrum donghaiense Lu ranged 20-27 ℃, thalassiosira sp. of 15-21 ℃, skeletonema costatum of 20-25 ℃, alexandrium tamarense of 17-25 ℃, and pyramidomonas delicatula of 24-28 ℃ (He et al., 2007; Chin et al., 1965). Salinity influenced penetration pressure of alga, thus physiological state would be changed to some extent. Each group of phytoplankton had suitable specific range of the salinity, e.g. prorocentrum donghaiense Lu prefers 25-35 psu (He et al., 2007). Therefore, 3-D baroclinic fields were significantly important for the modeling for biochemical cycling.

Time series observations of sea surface temperature and salinity were collected in May 2019 to verify the simulation (Figs. 4 and 5). Temperature showed a typical gradual increase pattern in late spring month, with higher values in southern domain. The maximum RMS error between the observed and the modeled was 1.39 psu. Salinity showed a strong signal of tides influence on flood/ebb cycle.

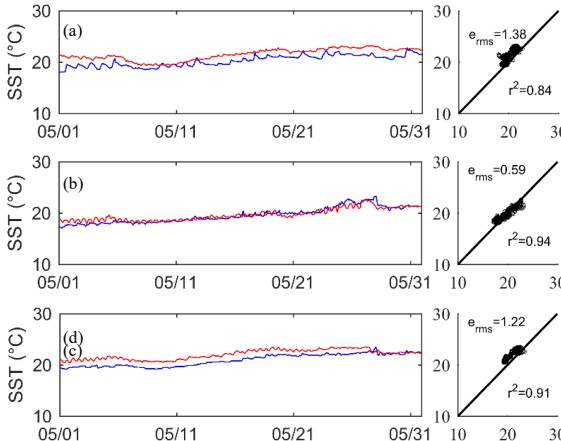


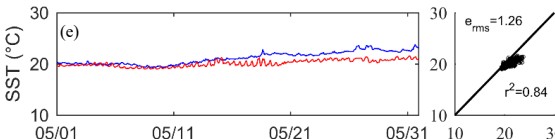

**Fig. 4. Validations against hourly water surface temperature during May 2019 at six tidal stations: KM (a), RA (b), SS2 (d), and PT (e). (blue line for the observed and red line for the modeled).**


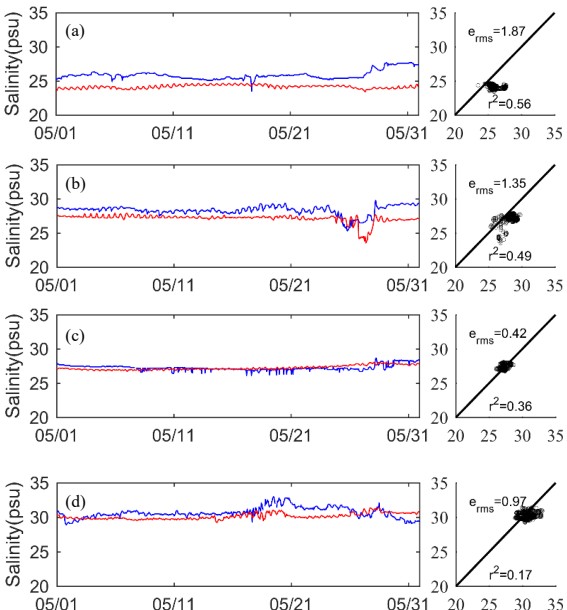

**Fig. 5. The same as Fig.4 except for surface salinity.**

**4.2 Biochemical fields**

Compared to physical oceanography, field observation for chemical and biological oceanography was scarce and this remained an obstacle to improve coupled biogeochemical model system (Ward et al., 2010). The Chla was commonly collected in the estimation of coupled biogeochemical model for wide availability of the data, both in-situ and remotely sensed, and was a focus of this model assessment.

Validation for nutrients and Chla at observed sites were also collected.

**4.2.1 Nutrients**

Time series for simulated nutrient concentrations at four stations were shown in Figs. 6-8. Measured ammonium indicated large values near the coastal zone, e.g. wz02 and wz01, while it fluctuated over 1.0 mmol m$^{-3}$ of offshore station (Fig.7). Nitrate concentrations generally showed higher values in





offshore region, about twice than simulated values (Fig.8). The simulated phosphate was at good

agreement with the observations (Fig. 6). Generally, a reasonable agreement for nutrients was achieved

in terms of concentration magnitude. The wz01, located near the coastline of the Leqing Bay, was

affected by pollutants input. The model underestimated the observed NO3 at this site, which

approximately equaled to the initial fields.

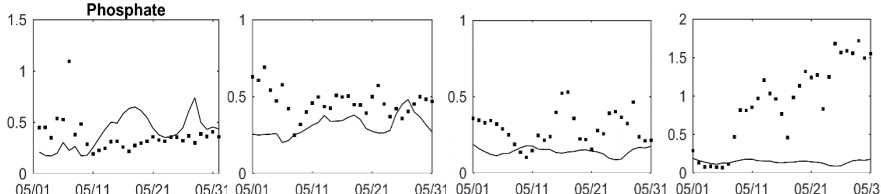

**Fig. 6. Validations of model results for temporal variation of Chla against the observations at nb03, zs03, wz02 and wz01, respectively (Data were marked by the squares and the modeled by solid lines) (unit: mmol m-3).**

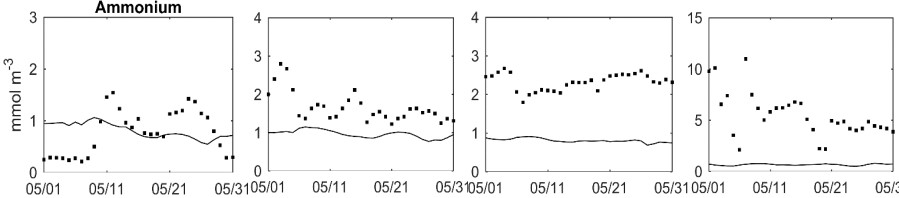

**Fig. 7. The same as Fig.6 except for the ammonium.**

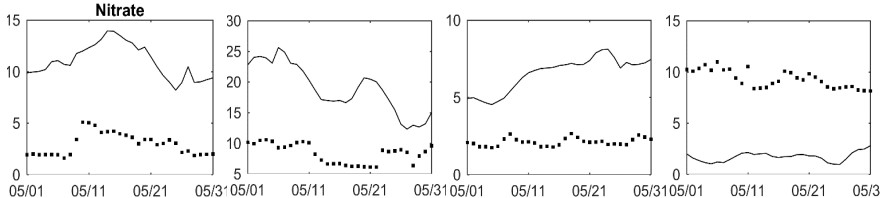

**Fig. 8. The same as Fig.6 except for the nitrate (unit: mmol m-3).**

**4.2.2 Chla and DO**

Figures 9 and 10 presented the simulated Chla and DO at four stations (nb03, zs03, wz01 and wz02).

The range of oxygen concentrations at each station were well reproduced, with some overestimation at

the wz01. Overall results suggested that oxygen budget in the system was satisfactorily achieved.

A comparison with Chla showed that the model has broadly captured observed variation, although with

lower magnitude of the nb03 and wz02, where bloom peak occurred in late-spring month. Field values

could reach to 15.0 mg Chla m$^{-3}$ at nb03, and almost 35.0 mg Chla m$^{-3}$ at wz02. Figure 11 showed the

comparison between modeled Chla and ocean color product of the Sea WiFS. Simulated Chla were in



typical average range of 0.0-20 mg Chla m$^{-3}$ with an overestimation near the YR offshore areas, and an

underestimation along the Zhejiang Province coast, e.g. OR estuarine areas.

Generally, the model was able to reproduce major temporal and spatial variability, although there was a

mismatch existed in late spring abundance. The reasons were possibly due to followings: (1) input values

of properties of coastal rivers used were not daily but monthly averaged, (2) inaccurate parameters for

chlorophyll synthesis, growth rates, etc., (3) absent of suspension of sediments and frequent high

turbidity in tidal estuaries along the coast.

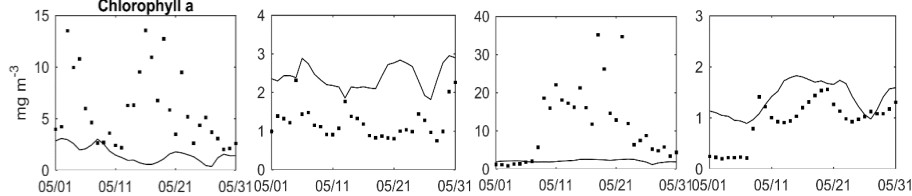

**Fig. 9. The same as Fig.6 except for surface Chla.**

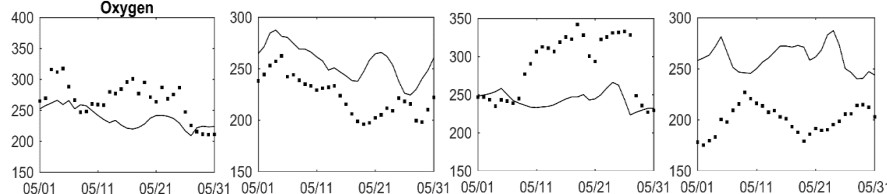

**Fig. 10. The same as Fig.6 except for the oxygen (unit: mmol m-3).**

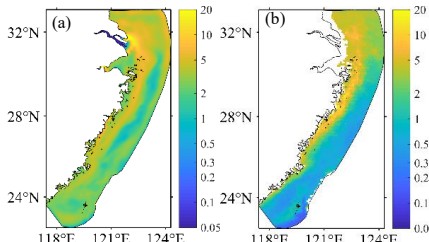

**Fig. 11. Mean simulated Chla compared to SeaWiFS ocean color products of May 2019.**

**4.2.3 Quantitative assessments**

Following statistical measures: the Cost Function (CF), the Percentage of Bias (PB) and the Adjusted

Relative Mean Absolute Error (ARMAE), were introduced to assess biological models usefulness as

tools in decision-making process. These statistics delivered model performance were defined as follows

(Tomasz et al., 2014):



$$CF = \frac{\sum |M - O|}{n\sigma_O} \qquad (3)$$

$$PB = \left| \frac{\sum(O-M)}{\sum O} \cdot 100 \right| \qquad (4)$$

$$ARMAE = \frac{\langle |M-O| - OE \rangle}{\langle |O| \rangle} \qquad (5)$$

where observations, $O$ and model $M$, fields were defined in a unstructured spatial grid and in time. $n$

represented the number of observations. $\bar{O}$ in above equations represented observation averages, $\sigma_O$

was the standard deviation of all observations and $OE$ was observational error. The angular brackets in

Eq. (5) denoted the averaged, and negative values in the numerator of this equation were set to zero

before averaging. A conservative estimate of the observational error was used at 50% (absolute relative

error). The performance indicator cited the three metrics: PB (<10 = excellent, 10-20 = very good, 20-

40=good, >40 = poor/bad), ARMAME (<0.2 = excellent/very good, 0.2-0.4 good, 0.4-0.7 = reasonable,

0.7-1.0 = poor and >1.0 = bad), and CF (<1 = excellent/very good, 1-2 = good, 2-3 = reasonable and >3

= poor/bad) (Tomasz et al., 2014).

The statistic metrics defined by Equations. (3)-(5) for the modeled and the observed were collected in

Table 6. The model was assessed by CF as excellent/very good for ammonium, phosphate and Chla, and

good for nitrate and DO. The values of ARMAE represented the relative error over and above the

estimated error in the observations. The model scores good for ammonium and Chla, excellent/very

good for phosphate and DO, and poor for nitrate. The assessment based on PB was more rigorous, as

PB showed the bias normalized by the observations rather than standard deviation. The model was

classified as excellent/very good for DO, and poor/bad for the remaining state variables.

**Table 6. Averaged statistical measures of model-observed comparison for the surface Chla, ammonium, nitrate, phosphate and DO of the four eco-buoys (highlighted for poor, and the other for good or excellent).**

|        | NH4   | NO3   | PO4   | Chla  | DO    |
|--------|-------|-------|-------|-------|-------|
| PB     | *67.8* | *65.4* | *53.6* | *60.1* | 14.2  |
| ARMAE  | 0.22  | *0.84* | 0.19  | 0.33  | 0.0   |
| CF     | 0.68  | 1.41  | 0.78  | 0.45  | 1.32  |





## 5 Analysis and discussion

### 5.1 Nutrients

Figure 12 showed modeled distribution of the nutrients in May of bloom peak month. Nutrients exerted the control to some extent on phytoplankton composition in some systems. It was particular relevant for silica because the model estimated a decrease of this nutrient from the inshore region to offshore region. Both nutrients showed a higher value near estuarine and nearshore areas, especially the YR estuary and coastal areas of Zhejiang Province. Affected by the interaction of the YR diluted water and longshore current, four type of nutrients roughly kept similar trend that decreased from the inshore plume region to offshore region. Nutrient front was formed offshore with different patterns. Nutrients were depleted in the well-mixed areas away from the front. The patterns appeared roughly same trend with observations in May 2015 (Ye et al., 2015), although the data were collected in a different year.

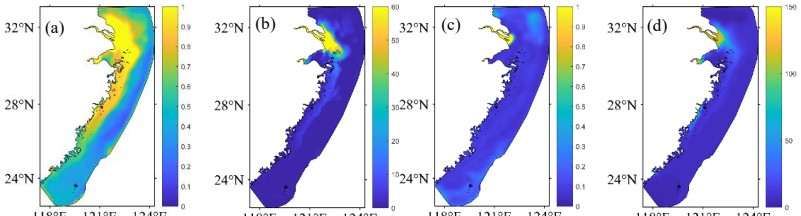

**Fig. 12. Model results: mean distribution of ammonium (a), nitrate (b), phosphate (c), and silicate (d) of May 2019.**

### 5.2 Phytoplankton

Current pelagic model of ERSEM comprised four functional types for primary producers based on single trait size, with the exception of the requirement of the silicate by diatoms and an implicit calcification potential of nanoflagellates. This lead to the four classes of diatoms, and pico-, nano-, and microphytoplankton (Butenschön et al., 2016). As shown in Fig. 13a, high values of diatoms appeared at the QT and OR river mouth, and mean value of 2-5 mg C m$^{-3}$ in plume region and offshore areas. The nanophytoplankton occupied almost entire domain except for upstream river mouth with relatively smaller values (Fig. 13b). For picophytoplankton, obvious higher values occured outside the YR mouth, where Guo et al. (2014) also concluded the similar pattern according to two curises in August 2009 (Fig. 13c).

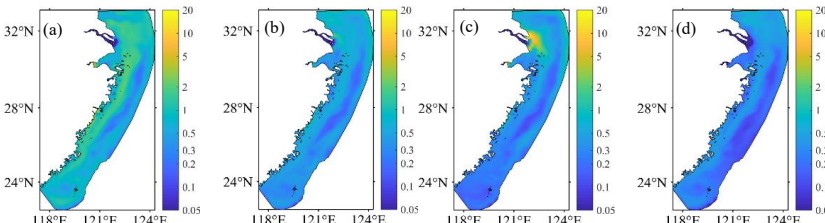

**Fig. 13. Mean distributions of phytoplankton of May 2019 (units: mg C m-3): diatoms (a), Nanophytoplankton (b),**
**Picophytoplankton (c), and Microphytoplankton (d), respectively.**

**5.3 Zooplankton**

Three predators were considered including the microzooplankton, mesozooplankton and heterotrophic

flagellates according to their size classes. They predated on different prey types, including cannibalism

(Butenschön et al., 2016). Grazing was treated as major control on phytoplankton abundance in pelagic

system, but the lack of field data on trophic structure in coastal areas limited assessing the magnitude of

this control in coastal ECS. Observational data were not available for the model, moreover, zooplankton

research was outside the aim of our present study, so we only plotted spatial distributions of the three

zooplankton for the integrity and made some rough estimates (Fig. 14). It showed clearly that

heterotrophic flagellates occurred mainly outside YR estuary and along the coast of ZP. High values of

microplankton appeared near offshore plume areas, while the mesozooplankton grew along the coast

and high value could be seen in offshore plume region.

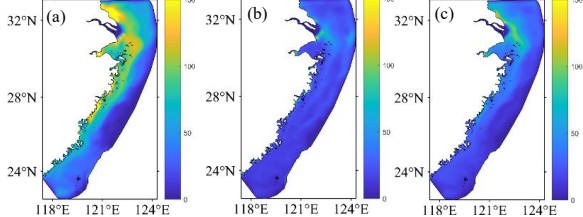

**Fig. 14. Mean distributions of the zooplankton of May 2019 (units: mg C m$^{-3}$): heterotrophic flagellates (a),**
**microzooplankton (b), and mesozooplankton (c), respectively.**

**5.4 Affecting factors on model accuracy**

There are several aspects affecting the accuracy of FVCOM-ERSEM coupled simulations of coastal

ECS, such as river inputs, open boundary conditions, model parameterization, and initial conditions.

Along coastal ECS, several significant rivers empty the open sea, e.g. YR, QR, OR, FR, MR and JR.

Apart from these runoff, there are many small rivers or streams, which affected circulation, especially


water quality near river mouths. Unfortunately, most of those small rivers are not continuously measured. Biological state variables of rivers including nutrients, DO and biomass, are much more difficult to be collected, which usually are assumed with historical data or references. Accurate field data of river mouths would improve the accuracy of model simulation greatly, particularly around estuarine areas.

Model results are dependent on the parameterizations of physical and biological processes in component

modules of FVCOM-ERSEM. For example, number and type of nutrients influenced biological results. Adsorption of suspended sediment and resuspension near bed affect the nutrient cycle. Also the dynamics of algal groups determined lower trophic food web.

Compared the above factors, open boundary conditions and initial conditions are the other potential factors. Although physical forcing is good enough, biological data are significant insufficient at open

boundary. The resolutions of initial salinity and temperature are important to circulation and biogeochemical model results.

**6 Conclusions and future work**

This paper presented a 3-D finite-volume physical-biogeochemical coupled model of coastal ECS. We implemented the 3-D baroclinic physical model FVCOM, which utilized a triangular horizontal grid to

better fit estuarine and coastal geometry, coupling with ERSEM, a well-established ecosystem model for lower trophic levels of marine food web, through the framework of FABM.

The model performance was assessed by extensive validation for major characteristics both in physical and biological environments, including the variables of water elevation, temperature, salinity, surface Chla and nutrients concentrations. Due to the limitation of observational data, we evaluated simulated

results of phytoplankton on a qualitative basis. The model was capable to reproduce main observed temporal and spatial features for phytoplankton and nutrients. The nutrients and phytoplankton distributions of coastal region of the ECS were discussed briefly in this study.

This integrated ecosystem model could be further explored to assess the response of biogeochemical process to physical, chemical and environmental parameters for coastal ECS. Certainly, we will continue

to improve the physical model and biogeochemistry model parameter space.





**Acknowledgements**

This work was supported by the Ministry of Science and Technology of China through the National Key
Research and Development Program (Grant number 2016YFC1401802), and supported by the State
Ocean Administration of China through the Ocean Public Science and Technology Research Funds
Projects (Grant number 201305012). We thank the research group in Plymouth Marine Laboratory for
sharing the code. We also thank Zhumei Che and Congjiao Zhao for explanations of in situ data.

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
