# Peer review of "Implementing a finite-volume coupled physical-biogeochemical model to the coastal East China Sea"

_Ocean Science, 2020_

## Referee Comment (RC1) · Anonymous Referee #1 · 24 Jul 2020

The authors presented their work on coupling a hydrodynamic model (FVCOM) and an ecosystem model (ERSEM), and the first results from model application to the coastal East China Sea.

Such kind of effort is doubtlessly important for understanding and quantifying biogeochemical processes in coastal shelf seas. However, the drawback of this study is obvious. It reads like an immature technical report rather than a scientific paper. It lacks a specific scientific topic to address, and I don't see any innovations in the presented results.

Further, the model setup for the study area is too simple to account for the complex

physical and biogeochemical processes that act in the East China Sea. The oversimplified treatment of the open boundary, which considers astronomical tides only, makes the model incapable of including the impact of the Kuroshio current that is so dominant over the outer and mid-shelf and even parts of shallow coastal area that is subject to intrusion of the Kuroshio current where a remakable front is formed (see e.g. Hsueh, 2000. The Kuroshio in the East China Sea. JMS 24, 131-139). Without an inclusion of the Kuroshio current, any numerical model for the East China Sea is deficient in accounting for the complexity of hydrodynamics, not to mention biogeochemical parameters that are highly depending on the interaction between the Kuroshio currents, coastal currents and river runoffs. The big gaps between measured data and model results shown in Fig. 6-10 clearly demonstrate the model deficiency.

Given that the drawback of the study (i.e. lack of a specific scientific question) and the model setup (i.e. exclusion of the Kuroshio currents) is so remarkable, I recommend to reject the manuscript in its current form.

---

## Author Comment (AC1) · 27 Jul 2020

Thanks very much for the significant precious comments from anonymous referee #1.

The main aims of this manuscript are to present the coupled finite-volume FVCOM with ERSEM, and to show preliminary qualitative analysis. We will explore and address specific scientific question (e.g. the response of biogeochemical parameters on certain physical processes, and quantitative analysis) in the following research (other manuscript). With the above arrangement, we admit this manuscript shows drawback of very specific questions.

[Figure]

We originally thought ocean circulation (e.g. Taiwan Warm Current and Kuroshio) may influence the coastal area little within 180-m isobath, which could be ignored. Actually, ocean circulation plays a role on the biogeochemical process. Thus, as the comments of referee #1, this drawback is obvious. We are improving open boundary conditions of the coupled model with ocean circulaitons.
* * *

---

## Referee Comment (RC2) · Anonymous Referee #2 · 28 Aug 2020

Here, I try to provide my comments about this manuscript. In general, the topic is appropriate for this journal, and results about the application of the FVCOM-ERSEM modelling system for East China Sea is very interesting.

The used tools are state-of-the-art and appropriate.

However I think that the article cannot be published without any changes, which I try to explain within the following lines.

Major comments:

- From my understanding, the authors want to present their model system and the

work they have done to set it up. I can understand that, as usually there is quite a lot of work to come to this point. However, the paper could be a bit more substantial if the authors could investigate some research question apart from demonstrating that their modelling system is doing realistic but fundamental things. Again, it is really some work to get the system running and collecting all the forcing data and so on. But, from my personal perspective, the article might win a lot, if some question could also be answered.

- The authors should mention some other modelling studies in that region. From a quick literature search I could find at least two similar studies, which are not mentioned by the authors: L. Zhao and X. Guo (2011) (Influence of cross-shelf water transport on nutrients and phytoplankton in the East China Sea: a model study, Ocean Sci., 7, 27–43, 2011) and Liu et al. (2010) (Seasonal variation of primary productivity in the East China Sea: A numerical study based on coupled physical-biogeochemical model, Deep-Sea Research II 57 (2010) 1762–1782). Perhaps the authors could say some words how their work might be related to these studies or other modelling studies and if and why there model system might be a progression.

- When reading my review, you will see that I'm not an English native speaker. So, I know how difficult it is to write in a foreign language. Reading this manuscript at hand, I got the impression that it might be helpful to get the English writing and formulations checked by some native speaker or a proofreading service. You will find examples of small errors below. I made good experiences with proofreading services, as my writing is also far from perfect.

- From perspective, the biological validation is a bit problematic. The authors mentioned some reasons for the mismatch as lacking river data or initial and boundary conditions. Is this really the total story? Perhaps the authors can say some more words why the biogeochemical module does not capture the data.

Some minor comments:

- Why do you focus on one month in May 2019. Actually, it might be more helpful to see an annual time frame for model validation. There is data around in other years for the ECS. You have modeled eight months, why not showing it to see some seasonal aspect.

- Line 74: What do you mean by 'providing environmental drivers offline'?

-Line 75: Please make it more clear what you mean with 'Online three-dimensional ...'

- Line 126: Please write some more words, what FABM is actually doing in this coupled modelling system

- Line 149 ff. Please provide some more references to you data sources. Usually, there are publications that can also be cited apart from simple internet sites.

- Line 234 and others: The unit of salinity should not be psu. Use g/kg or ‰ instead.

- Line 256: You should say reasonable with respect to the order of the nutrient concentration.

- Line 278: I think this sentence is to optimistic. The temporal variability is not kept from my respective very well. Please say some more words, when and where the model captures the data in a good way; and also when not.

- Table 6: The measures PB and CF show some opposite behaviour. Why? And what does this mean?

- Line 380: The last sentence is too optimistic from my perspective. Please show, that your deviations are comparable to other modelling studies or write it more appropriatly.

Some direct comments to the text:

- Abstract, line 12: Please let this sentence be checked by a native speaker. For me it sounds a bit strange: . . . has been implemented to a high . . . .

- Abstract, line 16: The authors are claiming to have done an extensive study of the

influence of different parameters on the biogeochemistry of the ECS. Please make this more clearly in the text, where these parameter investigations are done.

- Line 56: I think it might be 'model for ...'

- Line 64: I think it might be 'In recent years, ...'

- Line 73: Please find another word for 'defects'

-Line 84: 117-124.5

- Line 88-89: Please make the writing of the rivers consistent; e.g. YR or YR river?

- Line 109: You should make wave more specific; wind-generated surface gravity waves

- Line 125: . . . heterotrophic . . .

- Line 138: 102,688, (you mean non-overlapping?), 53,512 (no space between the comma)

- Line 141: . . . constitutes . . .

- Line 155: yearly-averaged (no space)

- Line 156: nitrate nitrogen

- Fig 2: The panels are to close to each other, please make some more space. The legend in panel b) does not fit to names in the caption.

- Table 1: It might be Si and not SI

- Table 2: Use Michaelis-Menten consistently (MM), Maximum N:C (not Nitrogen:C)

- Line 179: I think, presence might be right: concerns . . ..

- Line 186: modeled and observed what ???

- Line 189: modeled and observed what ???

- Line 196: modeled and observed what ???

- Line 202: Fig. 3 or Fig.3 ??

- Line 205: What is an R station

- Line 208: Please provide a more detailed refernce to T-tide tool kit

- Fig. 3: Please provide the axis-description in the scatter plots. What is modelled data and what observed data. The b) is misplaced. Please reduce the range of the SSH axis to -3 . . . 3 m (or -3.5 . . . 3.5), to see the difference more clearly.

- Table 5: Which line corresponds to the modeled, which to the measured data. Perhaps you want to separate the stations by horizontal lines?

- Line: 223: . . . structure is . . .

-Line 223-224: I think you want to mention the optimal temperature range, don't you?

- Line 229: Instead of using baroclinic fields, perhaps say temperature and salinity distributions

- Fig. 4: Please annotate the axes in the scatter plots. Please reduce the range of the temperature axis to see the difference more clearly.

- Line 247: you mean your bioge. model system (ERSEM) or these models in general?

- Line 250: Please rewrite this sentence. Validation data . . . were collected????

- Fig. 6: The caption says Chla, the heading is phosphate ????

- Line 273: Fig. 11 shows . . .???

- Fig. 11: Please write wich panel is SeaWiFS, which is model.

- Line 297: There is no bar(O) in the equations above.

- Line 317: . . . shows . . .

- Line 375: I think 'coupled'

- Line 379: . . . and nutrient concentrations ...

---

## Author Comment (AC2) · 2 Sep 2020

We also collected some more river data, especially of water quality, in order to correct river discharges. We will redo the validation by new results considering the Kuroshio Current and Taiwan Warm Current, and improved river input. Harmful algal blooms (HABs) outbreaks through later spring to early summer (from April to early June) along coastal ECS. In following version, we will investigate daily variations of nutrients and chlorophyll-a, and will explore the environmental impacts on HABs, especially the fate of diatom and dinoflagellate from early April to later May.

---

## Author Comment (AC3) · 7 Sep 2020

Thanks a lot for excellent job from anonymous referee #2, who has provided constructive suggestions and made amount of good revising opinions. We are improving the coupled model as comments from referee #1 with considering the intrusion of the Kuroshio Current and Taiwan Warm Current at open boundary. Also, we collected some more river data, especially of water quality, in order to correct river discharges. Harmful algal blooms (HABs) outbreaks through later spring to early summer (from April to early June) along coastal ECS. Additionally, we have collected ecological data of buoys in May. Thus, we validated the coupled model and showed some results in May. In follow-

ing version, we will investigate daily variations of nutrients and chlorophyll-a, and will explore the environmental impacts on HABs, especially the fate of diatom and dinoflagellate from early April to later May. We will mention some more modelling studies in coastal ECS, including the literatures advised by referee #2, and give our progressions. As for the biological validation, we will redo it with new results considering sea current and more river data. The writing was corrected carefully by one author of Qingqing Pan, who wrote PhD thesis in English and received her PhD degree in Norway. I think it is still not as good enough as that from native authors. We really appreciate those minor comments, which are greatly important for us. We will correct them.

---

## Author Comment (AC4) · 7 Sep 2020

Seasonal variation, especially along Zhejiang Province coast, is of great significance. We shall do some work on this aspect, since previous studies focused on shelf-scales or limited observations. As mentioned above, we collected data of ecological buoys and HABs information in May, thus we demonstrated the validations and results in May. We may add buoys data in April, which was collected very recently. Harmful algal blooms (HABs) outbreaks through later spring to early summer (from April to early June) along coastal ECS, therefore, studies during this period is of more significance rather than other seasons.

---

## Author Comment (AC5) · 9 Sep 2020

We will also redo "Fig. 11 mean simulated Chla compared to SeaWiFS ..." with OC_CCI data in 2019, since the former is climate monthly-averaged data and not good enough. Previous studies and observations have shown that diatom blooms in April and dinoflagellate outbreaks in May. Therefore, variations of nutrients and biomass in April will be analyzed as supplement, in order to explore diatoms and dinoflagellate processes.